# Maternal Choline Supplementation Modulates Placental Markers of Inflammation, Angiogenesis, and Apoptosis in a Mouse Model of Placental Insufficiency

**DOI:** 10.3390/nu11020374

**Published:** 2019-02-12

**Authors:** Julia H. King, Sze Ting (Cecilia) Kwan, Jian Yan, Xinyin Jiang, Vladislav G. Fomin, Samantha P. Levine, Emily Wei, Mark S. Roberson, Marie A. Caudill

**Affiliations:** 1Division of Nutritional Sciences, Cornell University, Ithaca, NY 14850, USA; jhk288@cornell.edu (J.H.K.); sk2563@cornell.edu (S.T.K.); jy435@cornell.edu (J.Y.); XinyinJiang@brooklyn.cuny.edu (X.J.); vgf6@cornell.edu (V.G.F.); spl63@cornell.edu (S.P.L.); ew376@cornell.edu (E.W.); 2Department of Health and Nutrition Sciences, Brooklyn College, Brooklyn, NY 11210, USA; 3Department of Biomedical Sciences, Cornell University, Ithaca, NY 14850, USA

**Keywords:** choline, Dlx3, placenta, placental insufficiency, inflammation, angiogenesis, apoptosis

## Abstract

*Dlx3* (distal-less homeobox 3) haploinsufficiency in mice has been shown to result in restricted fetal growth and placental defects. We previously showed that maternal choline supplementation (4X versus 1X choline) in the *Dlx3*+/− mouse increased fetal and placental growth in mid-gestation. The current study sought to test the hypothesis that prenatal choline would modulate indicators of placenta function and development. Pregnant *Dlx3*+/− mice consuming 1X (control), 2X, or 4X choline from conception were sacrificed at embryonic (E) days E10.5, E12.5, E15.5, and E18.5, and placentas and embryos were harvested. Data were analyzed separately for each gestational day controlling for litter size, fetal genotype (except for models including only +/− pups), and fetal sex (except when data were stratified by this variable). 4X choline tended to increase (*p* < 0.1) placental labyrinth size at E10.5 and decrease (*p* < 0.05) placental apoptosis at E12.5. Choline supplementation decreased (*p* < 0.05) expression of pro-angiogenic genes *Eng* (E10.5, E12.5, and E15.5), and *Vegf* (E12.5, E15.5); and pro-inflammatory genes *Il1b* (at E15.5 and 18.5), *Tnfα* (at E12.5) and *Nfκb* (at E15.5) in a fetal sex-dependent manner. These findings provide support for a modulatory effect of maternal choline supplementation on biomarkers of placental function and development in a mouse model of placental insufficiency.

## 1. Introduction

Abnormal placental development underlies many pathologies of pregnancy including preeclampsia, intrauterine growth restriction (IUGR), and spontaneous abortion [1]. These conditions have serious consequences for the mother and fetus, and few treatments are currently available for their prevention or treatment.

Although the etiology of placental insufficiency is not fully understood, an imbalance of pro- and anti-angiogenic and inflammatory factors may contribute to it [1,2]. The development of the placenta requires extensive angiogenesis and vasculogenesis, and the maternal uterine spiral arteries must be invaded and remodeled by extravillous trophoblasts to allow for increased blood flow to the fetus [3]. This process requires adequate expression of angiogenic and remodeling genes including the vascular endothelial growth factor (VEGF) family. Inadequate trophoblast invasion is a characteristic of many placental pathologies. When this process is incomplete, blood flow to the placenta is compromised, and oxygen supply may be sporadic, leading to placental injury from hypoxia/reoxygenation. Oxidative stress, excessive apoptosis, and inflammation result, compromising trophoblast function and preventing efficient transfer of nutrients [4]. Anti-angiogenic factors in the maternal circulation, including soluble fms-like tyrosine kinase-1 (sFlt-1) and soluble endoglin (sEng), may contribute to the pathogenesis of preeclampsia symptoms including hypertension and proteinuria, and can be used as biomarkers or predictors of risk when measured in early gestation [5,6]. Together, these dysregulated processes interact to contribute to the pathogenesis of placental disorders such as preeclampsia and IUGR.

The essential nutrient choline is required for the synthesis of the neurotransmitter acetylcholine, the membrane phospholipid phosphatidylcholine, and the methyl donor and osmolyte betaine [7]. These biomolecules have crucial roles in supporting pregnancy via their effects on cell growth, DNA methylation, and signaling processes [8]. Choline has previously been shown to reduce levels of sFlt-1 in a randomized dietary intervention trial of maternal choline supplementation during the third trimester of pregnancy in healthy women [9]. Further, choline deficiency has been shown to increase inflammation and apoptotic processes in a cell culture model of human placental trophoblasts [10]; meanwhile, supplementing the maternal diet with additional choline decreases biomarkers of both processes in wildtype mouse placenta [11]. We, thus, hypothesized that choline supplementation would be beneficial for pregnancies complicated by placental insufficiency.

*Dlx3* (distal-less homeobox 3) haploinsufficiency in mice has been shown to result in restricted fetal growth and placental defects [12]. This homeodomain-containing transcription factor is required for the development of the maternal–fetal interface [13]. Placentas lacking one copy of this gene display inadequate vascularization and abnormal development of the placental labyrinth [12], which is the area of nutrient exchange between the mother and fetus. Heterozygous embryos are viable but their placentas display abnormalities including impaired remodeling of maternal spiral arteries as well as increased placental oxidative stress and apoptosis [12]. In mice that are homozygous null for *Dlx3*, placental failure occurs between embryonic days 9.5–12.5 resulting in extremely restricted fetal growth and subsequently death [13]. Notably, we have previously shown that maternal choline supplementation (4X versus 1X choline) in *Dlx3*+/− dams increases fetal weights during mid-gestation in wildtype, heterozygous, and null embryos [14]. Most important to this body of work is the evidence supporting a role of DLX3 in human placental insufficiency (reviewed in Reference [15]).

In the present study, we sought to use the *Dlx3*+/− mouse model of placental insufficiency to investigate the effects of maternal choline supplementation on biomarkers of placental function and development across gestation. To accomplish these aims, we used tissue samples that were previously collected from *Dlx3*+/− pregnant dams randomized to consume one of three levels of choline intake [14].

## 2. Materials and Methods

### 2.1. Mice and Diets

All animal protocols and procedures used in this study were approved by the Institutional Animal Care and Use Committees at Cornell University, and were conducted in accordance with the Guide for the Care and Use of Laboratory Animals. *Dlx3*+/− mice were genotyped using tail DNA with three-primer duplex PCR (Appendix A). Primers were designed to amplify a wildtype band, a knockout band, or both (indicating a heterozygote). Mice were housed in microisolator cages (Ancare, Bellmore, NY, USA) in an environmentally-controlled room (22–25 °C and 70% humidity) with a 12-hour light–dark cycle. During breeding, mice were given ad libitum access to commercial rodent chow (Teklad, Madison, WI, USA) and water. *Dlx3*+/− females were mated with *Dlx3*+/− males in a pair-wise manner, and their offspring were genotyped at time of weaning (3 weeks of age). Heterozygous pups were given ad libitum access to the 1X choline control diet containing 1.4 g choline chloride/kg (Dyets #103345). *Dlx3*+/− female mice were then randomized five days before mating with *Dlx3*+/− male mice and received either the 1X choline control diet, the 2X choline diet containing 2.8 g choline chloride/kg (Dyets #103346), or the 4X choline diet containing 5.6 g choline chloride/kg (Dyets #103347); all provided ad libitum. Embryonic (E) day 0.5 was designated via the presence of a vaginal plug. Pregnant mice were euthanized via carbon dioxide asphyxiation at four different gestational time points: E10.5, E12.5, E15.5, and E18.5.

### 2.2. Tissue Collection and Processing

After pregnant dams were sacrificed, embryos and placentas were carefully dissected with minimal decidual contamination and weighed. For approximately 1/3 of the litter, the implantation site was fixed in 10% formalin for histological analyses after removal of the embryo for genotyping. For the remaining placentas, at gestational time points E12.5, E15.5, and E18.5, placentas were bisected across the chorionic plate; one half was stabilized in RNAlater for mRNA analysis, while the remaining half was immediately frozen in liquid nitrogen and stored at −80 °C for metabolite analysis. Due to the smaller tissue size, E10.5 placentas were designated for mRNA analysis or metabolite analysis, alternately. Fetal DNA was extracted for *Dlx3* genotyping and sex determination using a commercial kit (Qiagen Inc., Germantown, MD, USA). Sex genotyping was performed using PCR for the *Sry* gene with a commercial kit (Qiagen Inc., Germantown, MD, USA). Primers are listed in Appendix A.

### 2.3. Quantitative Real-Time RT-PCR

RNA was extracted from placentas maintained in RNAlater using TRIzol reagent (Invitrogen, Waltham, MA, USA). Two to three *Dlx3* heterozygous placentas per dam were randomly selected for extraction. RNA concentration and quality were assessed with a NanoDrop ND-1000 instrument (Thermo Fisher Scientific, Waltham, MA, USA), and samples with an A260/A280 ratio above 1.8 were used for quantification. Reverse transcription was performed using the ImProm-II Reverse Transcription System (Promega, Madison, WI, USA). Quantitative PCR was performed using SYBR^®^ Green in a Roche LightCycler480 (Roche, IN, USA). All primers for the targeted genes (*Vegfa*, *Pgf*, *Eng*, *Mmp14*, *Tnfα*, *NfκB*, *Il1b*) were designed using NCBI Primer-BLAST (Appendix A). Reaction conditions were as follows: 95 °C for 5 min, followed by 40 cycles for 15 s at 95 °C, 30 s annealing (see Appendix A for annealing temperatures), and 30 s at 72 °C. PCR product specificity was monitored using melting curve analysis at the end of the amplification cycles. Fold changes were calculated using the ΔCt method [16] normalized to the expression level of the housekeeping gene *Tbp* (TATA box binding protein), which has previously been shown to be stable in placental tissue [17], and in response to varying choline supply [18]. At E10.5, both wildtype and heterozygous placentas were used due to limited tissue availability, and *Dlx3* genotype was included in the statistical model. All qPCR analyses were performed in triplicate.

### 2.4. LC-MS/MS

Concentrations of acetylcholine were measured in the placenta using LC-MS/MS according to the method of Holm et al. [19] with modifications based on our equipment [20].

### 2.5. Placental Morphometry

Placental tissues fixed in 10% formalin were paraffin embedded and sectioned at 10 µm. Immunohistochemistry was performed on formalin-fixed sections as described previously [21]. For the analysis of maternal spiral artery areas, placental sections were incubated with smooth muscle actin (SMA) antibody (1:50, DakoCytomatin, Glostrup, Denmark), followed by a secondary antibody. Slides were imaged using an Aperio Scanscope (Vista, CA, USA). Maternal spiral arteries were manually defined based on the staining location and the presence of non-nucleated red blood cells. Aperio ImageScope software, version 102.0.7.5, was used to quantify the area. Data are presented as the ratio of artery luminal area to total arterial area. For the analysis of the placental labyrinth area, placental sections were incubated with biotinylated GSL 1-isolectin B4 (1:100, Vector Laboratories, Burlingame, CA, USA) and 3-amino-9-ethylcarbazole (AEC; Invitrogen, Carlsbad, CA, USA), and counterstained with hematoxylin. Isolectin is a marker of endothelial cells and has been used previously to stain vasculature in other mouse tissues. The placental labyrinth compartment was defined manually based on staining location, and area was calculated using Aperio ImageScope software (Vista, CA, USA). Data are expressed as mm^2^ of cross-sectional labyrinth area.

### 2.6. Placental Apoptosis

Placental apoptosis was assessed using the terminal deoxynucleotidyl transferase dUTP nick end labeling (TUNEL) assay. A commercial kit (Millipore, Billerica, MA, USA) was used according to the manufacturer’s instructions. Sections were imaged using an Aperio ScanScope and the number of TUNEL-positive cells was determined in the decidua and labyrinth by the average number of TUNEL-positive cells in several randomly selected fields. Field sizes were as follows: for E10.5, five fields of 250 × 250 μm^2^; for E12.5, five fields of 350 × 350 μm^2^; and for E15.5 and E18.5, ten fields of 500 × 500 μm^2^.

### 2.7. Statistical Analysis

For all outcome variables, data were analyzed separately for each gestational day using a linear mixed model. All statistical models included choline treatment as a fixed effect, and a maternal identifier (ID) as a random effect. Litter size, fetal genotype (except for the gene expression outcomes at E12.5, E15.5, and E18.5, which only included heterozygous pups), and fetal sex (except when the data were stratified by fetal sex) were controlled for in the models as fixed effects. Normality of the data was assessed by evaluating the distribution of residuals. Due to the hypothesis-driven nature of the study and the relatively small sample sizes, corrections for multiple analyses were not performed and statistical interactions were not assessed. Data are presented as means ± SEM. *p* ≤ 0.05 was considered statistically significant and 0.05 < *p* < 0.10 was considered to indicate trends. All analyses were performed using SPSS software, Version 23 (IBM, Chicago, IL, USA).

## 3. Results

We previously reported that maternal choline supplementation increased placental and fetal weights at E10.5 in offspring of *Dlx3*+/− dams regardless of fetal genotype [14]. These data can also be found in Appendix A which depicts embryo weight, placental weight, crown rump length, placental efficiency, and litter size in *Dlx3*+/− dams in response to maternal choline intakes (1X control, 2X, and 4X) at E10.5, E12.5, E15.5, and E18.5.

### 3.1. Placental Morphometry

Because the *Dlx3* model is associated with labyrinth abnormalities, we sought to determine whether choline treatment increased placental weight by increasing the size of the labyrinth. At E10.5, 4X choline placentas had 73% larger labyrinth area versus 2X choline (*p* = 0.014), and tended to be ~46% larger than 1X control placentas (*p* = 0.092, Figure 1A and 1B). At E12.5 and E15.5, there were no significant differences in labyrinth size by choline treatment. At E18.5, 2X placental labyrinths were ~25% smaller compared to 1X controls (*p* = 0.037) and tended to be ~20% smaller than 4X choline placentas (*p* = 0.081, Figure 1A).

### 3.2. Placental Apoptosis

We performed the terminal deoxynucleotidyl transferase dUTP nick end labeling assay (TUNEL) to assess placental levels of apoptosis at all four gestational time points. At E10.5, choline treatment tended to result in ~49% lower TUNEL scores (average # TUNEL-positive cells per field) in the 4X choline placentas (*p* = 0.053 versus 1X controls) (Figure 2A). At E12.5, 4X choline placentas had ~58% lower TUNEL scores (*p* = 0.048 versus 1X controls) (Figure 2A,B). No differences in TUNEL score were detected between choline treatment groups at E15.5 and E18.5 (Figure 2A).

### 3.3. Placental Artery Remodeling

Because we had previously found an effect of maternal choline supplementation on spiral artery remodeling in wildtype mice [11], we investigated whether it would have a similar function in a model of placental insufficiency. We assessed the luminal ratio of maternal spiral arteries, defined as (luminal area)/total arterial area. Luminal ratio did not differ by choline treatment or *Dlx3* genotype at any gestational time point.

### 3.4. Placental Expression of Angiogenic Genes

Because altered levels of pro-angiogenic factors have been implicated in the development of preeclampsia and other pregnancy disorders, we measured mRNA expression of several key modulators of placental angiogenesis throughout pregnancy in *Dlx3*+/− placentas (Figure 3A–D). For this outcome, we stratified by fetal sex due to our previous findings showing effects of fetal sex on angiogenic gene expression [11].

At E10.5, expression of *Eng* was ~31% lower in the female 2X choline placentas versus 1X controls (*p* = 0.029) (Figure 3A). No effects of choline treatment on *Eng* expression were detected for the male placentas, nor did choline treatment alter *Vegfa*, *Pgf*, or *Mmp14* expression at E10.5 in either sex.

At E12.5, 2X and 4X choline males had lower expression of *Eng* vs. 1X controls (~64% and ~43%, respectively, *p* = 0.005, 0.013) (Figure 3B). Similarly, male 2X choline placentas had ~44% lower expression of *Vegfa* (*p* = 0.026 versus 1X) (Figure 3B). No significant effects of choline treatment on these angiogenic factors were detected in females; nor did choline treatment influence *Pgf* or *Mmp14* expression at E12.5.

At E15.5, expression of *Eng* in female placentas was ~38% lower in the 2X choline group (*p* = 0.002 vs. 1X). Similarly, 2X choline female placentas had ~17% lower expression of *Vegfa* vs. 1X controls (*p* = 0.031) (Figure 3C). Expression of *Mmp14* was ~42% lower in female 2X choline placentas (*p* = 0.019, vs. 1X) and ~41% lower in 4X choline placentas (*p* = 0.015, vs. 1X) (Figure 3C). No effects of choline treatment on these angiogenic factors were detected in males; nor did choline treatment influence *Pgf* expression in either fetal sex at E15.5.

At E18.5, male placentas in the 2X choline group had ~35% higher expression of *Vegfa* versus 1X controls and 4X choline groups (*p* = 0.011 and 0.01, respectively). Similarly, expression of *Pgf* tended to be ~29% higher in 2X choline males versus 1X control males (*p* = 0.091) and was ~42% higher versus 4X choline (*p* = 0.091). Male 2X choline placentas also had ~62% higher expression of *Mmp14* versus 1X placentas (*p* = 0.006) (Figure 3D). No effects of choline treatment on these angiogenic factors were detected in females; nor did *Eng* expression differ by choline treatment at E18.5 in either fetal sex.

### 3.5. Placental Expression of Inflammatory Genes

Abnormal regulation of inflammation has been shown in various placental pathologies; therefore, we also measured mRNA expression of several major inflammatory regulators in the *Dlx3*+/− placentas (Figure 4A–C). Again, data were stratified by fetal sex due to our previous work showing the strong effects of fetal sex on inflammatory processes [11].

At E10.5, there were no significant differences in *Tnfα*, *Nfκb*, or *Il1b* expression among choline treatment groups. At E12.5, 2X choline male placentas had a lower expression of Tnfα (~57% versus 1X) and tended to have lower expression of *Nfκb* (~50%, *p* = 0.096 versus 1X) (Figure 4A). No effects of choline treatment on these inflammatory genes were detected in females; nor did expression of *Il1b* differ by treatment.

At E15.5, female 2X choline placentas had ~33% lower expression of *Nfκb* versus 1X controls (*p* = 0.020). Expression of *Il1b* was ~36% lower in 2X choline placentas of both sexes versus 1X controls (*p* = 0.034, Figure 4B), and also in males alone (*p* = 0.031).

At E18.5, expression of *Il1b* was ~32% lower in 4X choline pups of male and female placentas combined (*p* = 0.02) (Figure 4C), a finding that did not reach statistically significant levels in either sex alone. Female 4X choline placentas tended to have ~39% lower expression of *Nfκb* versus 1X control females (*p* = 0.058) (Figure 4C). No differences in Tnfα expression were seen among choline groups at E15.5 or E18.5.

### 3.6. Placental Acetylcholine

Because acetylcholine has been linked to angiogenic and inflammatory signaling in the placenta [22,23], we measured concentrations of placental acetylcholine throughout gestation. At E12.5, 2X choline placentas had ~136% higher concentrations of acetylcholine versus 1X controls (*p* = 0.011), and ~107% higher concentrations versus 4X choline placentas (*p* = 0.020). At E10.5, E15.5, and E18.5, acetylcholine concentrations did not differ by choline treatment (Figure 5).

## 4. Discussion

### 4.1. Maternal Choline Supplementation in the Dlx3+/− Pregnant Dam Increases Placental Labyrinth Size and Decreases Placental Apoptosis in Mid-Gestation

Because we had previously reported a higher placental weight at E10.5 with maternal choline supplementation in *Dlx3*+/− dams [14], we sought to determine whether this effect could be explained by alterations in the size of the placental labyrinth or apoptosis levels. Development of the labyrinth region, which facilitates nutrient transfer between mother and fetus, is defective in *Dlx3*−/− animals [13]. At E10.5, a high dose of choline (4X the recommended intake) yielded a larger labyrinth size compared to the 2X choline dose, and tended to yield a larger labyrinth size compared to the 1X choline dose. Thus, an increase in surface area for nutrient transfer may have contributed to the larger embryo weights we previously reported in offspring (both +/− and +/+) of *Dlx3*+/− mothers whom received 4X choline supplementation [14].

An additional explanation for the larger placental weights could be through reductions in apoptosis. For example, 4X (versus 1X) choline placentas had ~58% lower TUNEL scores at E12.5, and a tendency for lower scores (49% lower) at E10.5. These findings concur with previous results showing reductions in apoptosis with choline supplementation in human trophoblast cells [10] and in placentas of wildtype mice [11]. Although apoptosis is required for normal placental development, elevated levels of apoptosis have been reported in preeclamptic and IUGR placentas [24,25,26]. Notably, the elevated levels of apoptosis in *Dlx3*+/− placentas had previously been rescued using a strong antioxidant, TEMPOL [12]. Therefore, a reduction in placental apoptosis in mid-gestation via maternal choline supplementation may have beneficial effects in reducing the risk of developing placental complications later in gestation.

In a preliminary histological examination of spiral artery remodeling, we did not find evidence that maternal choline supplementation altered arterial luminal area percentage in the *Dlx3*+/− mouse. This is in contrast to our previous findings in wildtype dams, where 4X choline significantly increased luminal area percentage at E10.5, E12.5, E15.5, and E18.5 [11]. It is possible that the pathological phenotype of spiral artery remodeling in the *Dlx3*+/− dam was too severe to be ameliorated by choline supplementation. However, due to the small size of our histology cohort, this finding should be confirmed in a larger study.

Although prior work has shown that *Dlx3* is required for normal placental morphogenesis (13) and reduced *Dlx3* gene dose results in elevated placental cell oxidative stress and apoptosis coincident with altered vascular remodeling (12), we did not detect any differences in labyrinth size, apoptosis levels, or spiral artery remodeling between *Dlx3*+/+ and *Dlx3*+/− placentas. We previously found that *Dlx3*+/− fetuses had comparable weights, but a higher expression of growth factor genes versus their *Dlx3*+/+ littermates when their mothers consumed a high-quality pregnancy diet [14], suggesting that the *Dlx3*+/− placenta may activate compensatory mechanisms that lead to results similar to wildtype placentas for certain outcomes.

### 4.2. Effects of Maternal Choline Supplementation on Placental Inflammatory and Angiogenic Gene Expression Vary by Gestational Time Point and Fetal Sex

Inflammation and angiogenesis have been shown to be dysregulated in pregnancy complications including preeclampsia, IUGR, and spontaneous abortion [1,27,28]. Because of this, we measured placental mRNA expression of major modulators of these processes to determine whether choline can influence their placental expression in the *Dlx3*+/− mouse.

Overall patterns seen included the tendency for pro-inflammatory and pro-angiogenic markers to be downregulated in response to choline supplementation, with the 2X dose frequently having a larger effect than that of 4X. For example, compared to 1X choline, 2X choline significantly reduced the expression of the proinflammatory factors *Tnfα*, *Nfκb*, and *Il1b* at various gestational time points of pregnancy; a finding that was not consistently observed with 4X choline. Previous studies have shown that choline deficiency increases the expression of pro-inflammatory cytokines in a human placental trophoblast cell culture model [10], and that maternal choline supplementation reduces inflammatory response in pregnant rats challenged with lipopolysaccharide [29]. The results seen in the current study with a 2X choline dose provide additional support for a role of choline in mitigating inflammation, and suggest that efforts to increase maternal choline intake may be a nutritional strategy to reduce placental inflammation and improve pregnancy outcomes especially since less than ten percent of U.S. pregnant women meet choline intake recommendations [30]. 

The weakening of the choline-induced anti-inflammatory effects with the 4X treatment may suggest variability whereby maternal choline supplementation has an anti-inflammatory effect up to a certain concentration, beyond which its effects are attenuated. Alternatively, a strong growth-promoting effect of 4X maternal choline supplementation was detected in these mice at E10.5 [14], which was followed by a gradual slowing of growth such that only minor differences were detected in fetal weights in late gestation (E18.5). Thus, the effects of 4X choline on inflammatory (and angiogenic) processes in the placenta may have been weakened by compensatory mechanisms engaged to prevent exaggerated fetal growth.

We also observed downregulation of placental expression of the proangiogenic factor endoglin with maternal choline supplementation at E10.5 and E15.5 in female placentas and at E12.5 in males. These effects were mirrored by similar downregulation of proangiogenic *Vegfa*, which has been shown to be elevated or decreased in preeclamptic patients depending on the study [31,32,33]. It is possible that the larger labyrinth observed at E10.5 in the current study, combined with the superior placenta efficiency observed previously in these mice [14], resulted in a decreased need for placental angiogenesis, leading to a downregulation of pro-angiogenic factors.

Endoglin also serves as the precursor to soluble VEGF receptor endoglin (sEng), an anti-angiogenic factor that prevents VEGF from acting on target tissues [5], and contributes to the development of placental dysfunction. In the current study, expression of *Mmp14*, the matrix metalloproteinase that cleaves endoglin into its pathogenic soluble form [34], was significantly lower at E15.5 in females with choline supplementation, which is suggestive of decreased generation of sENG. In males, however, placental expression of *Mmp14* was higher at E18.5 with 2X choline supplementation, suggesting greater generation of sENG. This putative increase in sENG in late gestation could have a beneficial role in preventing preterm birth by blunting the parturition promoting effects of VEGF signaling through PGF and VEGF [31,32], both of which also were upregulated at this time point.

The effects of maternal choline supplementation on placental gene expression were sex specific, as previously reported in wildtype mouse pregnancy [11]. Differential expression of angiogenic factors according to fetal sex has been reported in both normal [33] and preeclamptic [35] human pregnancies. Although the mechanisms remain unclear, sex-specific differences in epigenetic regulation and hormonal environment have been suggested to be contributing factors [36,37]. Our findings add to the body of evidence demonstrating that fetal sex should be accounted for when examining placental and fetal outcomes during pregnancy.

### 4.3. Placental Concentrations of Acetylcholine are Minimally Affected by Maternal Choline Supplementation

Because acetylcholine has been reported to play a role in angiogenic [23] and inflammatory [22] signaling in the placenta, we hypothesized that choline may be modulating gene expression by altering acetylcholine levels. Surprisingly, we did not detect an increase in placental acetylcholine concentrations in response to maternal choline supplementation at E10.5, E15.5, or E18.5. At E12.5, acetylcholine levels were higher in the 2X, but not 4X, choline group. The overall lack of effect of maternal choline supplementation on placental acetylcholine concentrations could reflect rapid uptake of the acetylcholine molecule by the developing fetus as we previously reported that 4X choline increased acetylcholine concentrations in the fetal brain of wildtype mice [38].

### 4.4. Strengths and Limitations

The study strengths include: (i) the use of a standardized diet and three choline intake levels; (ii) the inclusion of four gestational time points, which allows for a dynamic view of the changes that occur in the placenta throughout gestation; and (iii) the use of an animal model with a hemochorial structure similar to the primate placenta. Nonetheless, several limitations should be noted. The small sample size combined with the relatively high degree of variability for most of the outcome data precluded our ability to assess the interactive effects of fetal genotype and fetal sex with choline treatment. An additional limitation of the current study was the analyses of mRNA abundance without concurrent analyses of proteins. Finally, our histological analyses were included as a hypothesis-generating examination of choline’s effects in compromised pregnancies, and should be confirmed in larger cohorts.

## 5. Conclusions

In conclusion, maternal choline supplementation modulated placental development and function in a mouse model of placental insufficiency, providing some evidence that choline may be a useful therapy during pregnancy to prevent or overcome development of placental disorders that impact fetal and maternal health. Decreases in pro-inflammatory gene expression and placental apoptosis are especially encouraging and align with previous results showing similar effects of choline in different experimental models. Since many pregnant women do not currently meet the recommended intake levels of choline [30], further research is warranted to investigate the effects of increased maternal choline intake on human placental function.

## Figures and Tables

**Figure 1 nutrients-11-00374-f001:**
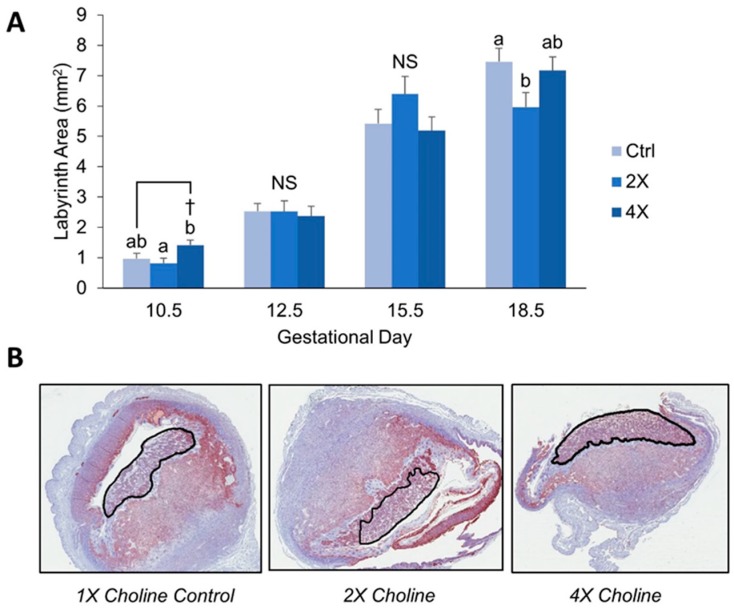
Placental labyrinth area at E10.5, E12.5, E15.5, and E18.5 by choline treatment (**A**). Representative images of placental labyrinths at E10.5 are shown in (**B**). Data were analyzed using mixed linear models with choline treatment as a fixed effect and maternal ID as a random effect. Litter size, fetal genotype, and fetal sex were controlled for in the models as fixed effects. Values are presented as mean ± SEM. Different letters denote significant differences between means of the treatments at *p* ≤ 0.05. † denotes *p* < 0.1. *n* = 7–12 placentas per treatment per time point. NS, nonsignificant.

**Figure 2 nutrients-11-00374-f002:**
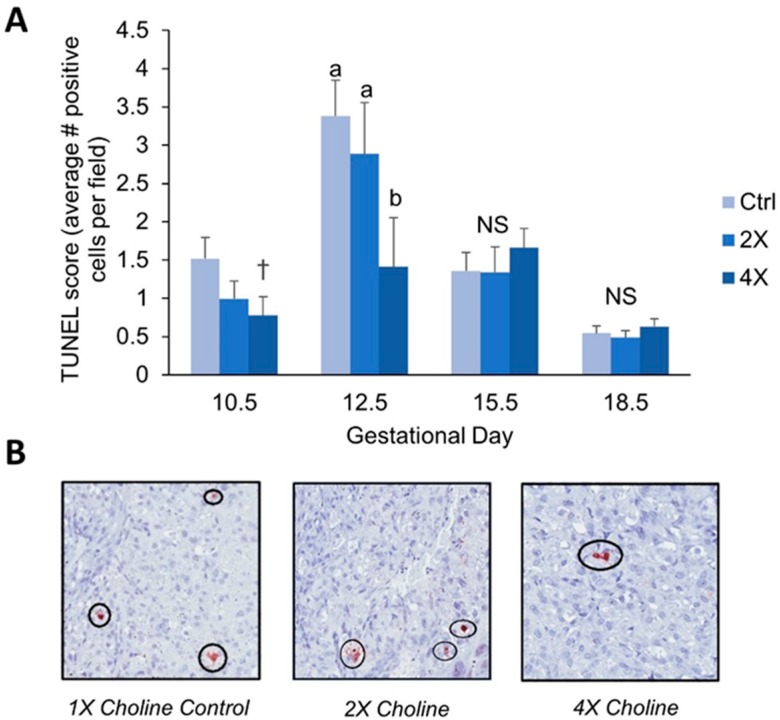
TUNEL score (calculated as the average number of TUNEL-positive cells per field) at E10.5, E12.5, E15.5, and E18.5 by maternal choline treatment (**A**). Representative images of TUNEL staining at E12.5 are shown in (**B**). Data were analyzed using mixed linear models with choline treatment as a fixed effect and maternal ID as a random effect. Litter size, fetal genotype, and fetal sex were controlled for in the models as fixed effects. Values are presented as mean ± SEM. Different letters denote significant differences between means of the treatments at *p* ≤ 0.05. † denotes *p* < 0.1 versus 1X choline controls. *n* = 7–15 placentas per treatment per time point. NS, nonsignificant.

**Figure 3 nutrients-11-00374-f003:**
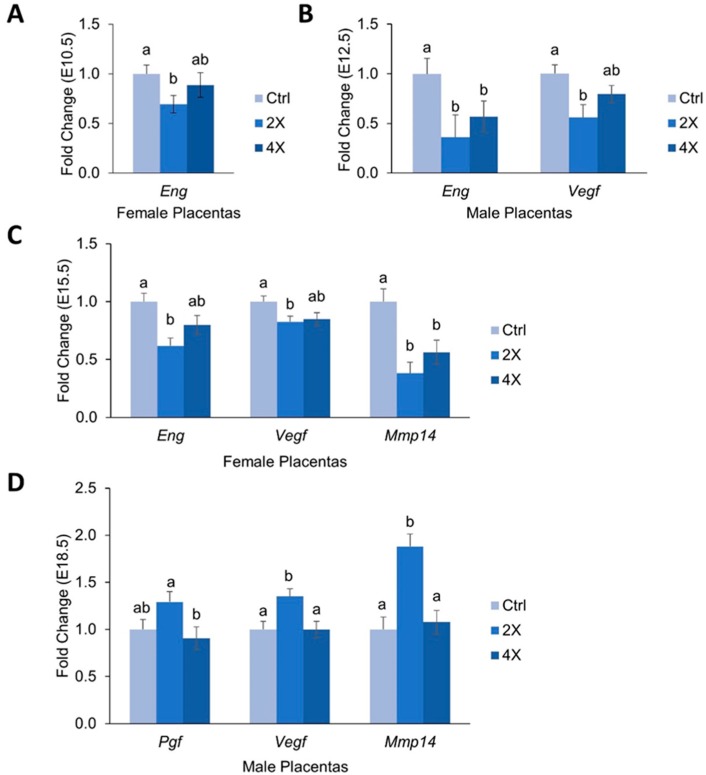
Placental mRNA abundance of angiogenic genes at (**A**) E10.5, (**B**) E12.5, (**C**) E15.5, and (**D**) E18.5 by maternal choline treatment. Data are expressed as the fold-change relative to the housekeeping gene *Tbp*. Data were analyzed using mixed linear models with choline treatment as a fixed effect and maternal ID as a random effect. Litter size and fetal genotype (for E10.5; heterozygous pups were used for E12.5, E15.5, and E18.5) were controlled for in the models as fixed effects. Values are presented as mean ± SEM. Different letters denote significant differences between means of the treatments at *p* ≤ 0.05. *n* = 20 placentas per treatment per time point.

**Figure 4 nutrients-11-00374-f004:**
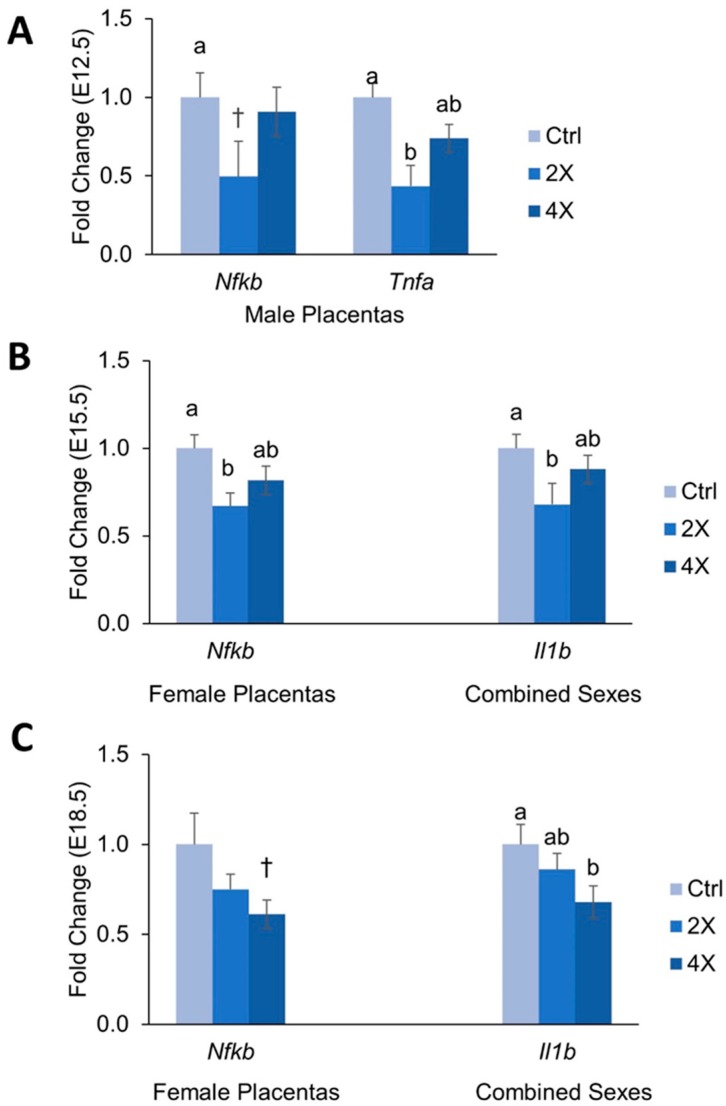
Placental mRNA abundance of inflammatory genes at (**A**) E12.5, (**B**) E15.5, and (**C**) E18.5 by maternal choline treatment. Data are expressed as the fold-change relative to the housekeeping gene *Tbp*. Data were analyzed using mixed linear models with choline treatment as a fixed effect and maternal ID as a random effect. Litter size and fetal genotype (for E10.5; heterozygous pups were used for E12.5, E15.5, and E18.5) were controlled for in the models as fixed effects. Values are presented as mean ± SEM. Different letters denote significant differences between means of the treatments at *p* ≤ 0.05. † denotes *p* < 0.1 versus 1X choline controls. *n* = 20 placentas per treatment per time point.

**Figure 5 nutrients-11-00374-f005:**
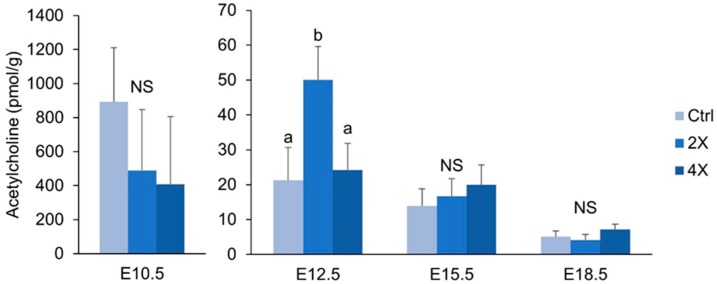
Placental acetylcholine concentrations (pmol/gram tissue) at E10.5, E12.5, 15.5, and 18.5 by maternal choline treatment (1X, 2X, and 4X choline). Data were analyzed using mixed linear models with choline treatment as a fixed effect and maternal ID as a random effect. Litter size, fetal genotype, and fetal sex were controlled for in the models as fixed effects. Values are presented as mean ± SEM. Different letters denote significant differences between means of the treatments at *p* ≤ 0.05. *n* = 20 placentas per treatment per time point. NS, nonsignificant.

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
