# Peer review of "Maternal Choline Supplementation Modulates Placental Markers of Inflammation, Angiogenesis, and Apoptosis in a Mouse Model of Placental Insufficiency"

_nutrients, 2019, doi:10.3390/nu11020374_

Reviewer 1 Report

Dear authors, this study was a lot of work and I want to thank you for doing it. Understanding of the effects of nutrition on pregnancy need to be moved out of the esoteric corner into the world of science, it’s high time. I’m hoping that my comments are not discouraging, they are not meant to be (if you feel discouraged, don’t forget that in general one always meets twice in life….). This is great work, in an ideal world these comments and suggestions make it better.

 Here we go:

-       The data depiction in the paper makes it a bit hard for the reader to follow along, I am referring to the bar graphs. While I do agree that it is very optically pleasing indeed! But science comes first in this case. I do request that the graphs in general are made much bigger for those of us of advanced age. I would much more prefer a graph style that depicts individual data points, a scatter blot of some sort. I understand that the graphs will be busy but the scatter blot will allow the reader to see both the distribution of the data and the n of each experiment which varies quite a bit between 7-20. It’s a big difference if we have 7 tightly clustered samples and that’s why the SEM is small or 20 wider spread samples with a very similar SEM. I would also recommend using Y axis in all blots and giving them ticks so one can use a ruler to identify numbers. For figures that have several graphs with a similar dimension such as the RNA analysis in figure 3, I would recommend using the same scale for all of them. 

-       There are letters “a”, “b”, and “ab” floating above the bars in the bar graph. I have a hunch what you’re trying to say but this isn’t working well. It’s confusing. First it took me forever to understand that that’s what you mean with the sentence “Differing letters denote….” And then you have cases like figure 1A right-hand side E18.5: ctrl and 4x are clearly not P<0.5 different but their letters indicate that they are.

-       The authors use a statistical model that I am not as familiar with as with other models, mixed linear models. That doesn’t mean at all that I am suggesting it is wrong by any means. However, for me to recapitulate the results of the calculations I will need to see the complete math for each of the figures and given that this is a more unusual model I would suggest including it in the paper at publication as a supplemental figure. I am also unclear why the strength of these models weren’t explored more fully? Did you alter which variables you set as fixed effects, did that change the outcome? Or maybe I’m missing something.

 -       Line 173: The authors state that “At E10.5, choline treatment tended to result in ~96% lower TUNEL scores…..” I must admit that I have trouble recapitulating that number. Looking at Figure 2A, the figure is a bit small and the Y axis doesn’t have any ticks but with a little bit of trying I estimated the TUNEL score for Ctrl to be ~ 1.5 and for 4x ~ 0.8. Even if I’m off a bit I still don’t see how the authors arrive at a 96% reduction. Please clarify.

 -       Figure 2: Is it possible to further specify in what region of the placenta the apoptotic cells were located? Morasso, Grinberg et al showed in PNAS in 1999 that the gene Dlx3 is expressed in several locations in the placenta, sub-locations would greatly contribute to the knowledge gain.

 -       Discussion point 4.1: For the points lines 272 to 289, please include the paper Morasso, Grindberg et al as well as other relevant publications. The lines 290 to 302 are enigmatic to me, I am not sure what the authors are referring to.

 -       In the introduction in line 56 the authors refer to a “randomized controlled feeding trial”. Given that this is a human study I would probably choose a different word than “feeding”. Just a suggestion.

 -       In line 57/58, the authors state that “choline has been shown to modulate….” Please explain in what way. Increase? Decrease? Turn red?

 -       In line 66 the authors state that the cited publication demonstrated, among other things, reduced invasion of trophoblasts. I had trouble finding that experiment in that cited publication, can you please clarify?

 -       Here are some thoughts of mine, just for your entertainment. The kinetics of your data shows nicely how the choline has its antiapoptotic protective mostly during time of highest growth/cell proliferation around day 12.5 and then it doesn’t do much with regard to apoptosis protection anymore but interestingly the consequences are not more labyrinth in the placentas but instead less inflammation. Whether that is causal or not is impossible to say but it is an interesting speculation because cell death is an immunostimulatory event and imbalance in that mechanism can wreak havoc in many ways. Also: it would be interesting to compare these numbers to dams that are Dlx3+/+ receiving choline. When Clark and Brown (Placenta, 2012) gave Dlx3+/- mice Tempol, a superoxide dismutase mimetic to alleviate oxidative stress, the placental labyrinth area of Dlx3+/+ mice pups also improved. In other words, sometimes even what we call our wildtype is not “ideal” and can be improved.

-       Lastly, this group of scientists obviously has a lot of experience with this model and has spent time thinking about what is happening here. I think the reader is very interested in the authors’ thoughts, for example on why the inflammation is reduced. Just a sentence or two. Tying maternal nutrition together with placental inflammation and morphological changes is something many of us are interested in. Or maybe draw a little model figure? Always highly appreciated! This work is bringing several loose ends together, great job!

 Author Response

A point by point response to Reviewer 1 has been uploaded as a PDF file.

Reviewer 2 Report

King et al present a study on the effects of choline supplementation on murine pregnancies associated with heterozygous loss of Dlx3. They suggest that dietary choline modulates placental development and function and expression of angiogenic and inflammatory gene expression in a model of placental insufficiency. Although the authors have elaborated on their previous study, this is a small step forward in understanding the effects of choline on Dlx3 haploinsufficiency in pregnancy.

 Major weaknesses:

-Why is Dlx3+/- the best model to use to investigate the effects of choline supplementation. A second model of ‘placental insufficiency’ should be compared?

-Much of the experiments seem underpowered, making overall interpretation of the findings of Dlx deficiency, choline supplementation and offspring sex difficult to interpret with any certainty.

-There is a lack of evidence presented on the relationship between the multiple effects of choline, and specifically its effects on methylation, and the function of Dlx3 as a transcription factor with a role in regulating trophoblast growth and inflammation.

 Further comments:

Abstract requires redrafting for the following reasons:

-the description of the Dlx+/- mouse placental phenotype is unclear: how is there placental insufficiency if fetal and placental growth both increase mid-gestation? The description of the phenotype could be described in better terms.

-the methodological part is over complicated, and results are unclear with regard to the strength of statements made according to the outcomes of the statistical analyses performed. 

-More affirmative statements should be made about the main outcomes and conclusion of the study.

 Introduction:

-2ndparagraph, sporadic blood supply would likely lead to injury from hypoxia/reoxygenation, rather than only hypoxia.

-The authors should include a paragraph to describe what is known about Dlx3 in human pregnancy.

 Methods:

-There should be more detail with regard to the section/sections used to estimate labyrinth area. A better analysis would be to use stereological techniques on 3 systematic random sections to estimate Lz volume as area estimates can be easily biased.

 Results:

-The results are difficult to interpret in the absence of data showing comparison between genotype exposed to different choline diets, despite the use of statistical modelling.

-Fetal and placental weights, F:P ratio, litter sizes and sex ratio should be presented.

-Given the interest in angiogenesis/vasculogenesis, a more complete analysis of the labyrinth zone should be included. Area of Lz does not provide sufficient detail as to the vascularisation of this zone.

-A more detailed analysis of apoptosis is required. Which cells were apoptotic?

-Expression of downstream targets of Dlx3 would be helpful, e.g. PpargGata2

-Given the effect of choline on methylation, were there difference in global methylation or methylation at binding sites of Dlx3?

Author Response

A point-by-point response to Reviewer 2's comments has been uploaded as a PDF file.

Reviewer 3 Report

The title should be corrected. None of the standard inflammation biomarkers were analyzed.

Line 61 It should be haploinsufficiency instead of insufficiency.

Lines 76-78 It is unclear why this sentence was added here.

Add information on diet intake control.

How many males and females were used for breeding?

Lines 89-93 This fragments seems unclear.

How many technical replicates of Q-PCR were performed?

Were there any differences in pregnant dams body weight?

Author Response

A point-by-point response to Reviewer 3's comments has been uploaded as a PDF file.

Round  2

Reviewer 2 Report

Although the authors have improved the manuscript's readability, the experimental design, and selected chosen techniques and the lack of power let the study down.